# From Prompts to Tokens: Internalizing Causal Supervision in Vision-Language Model for Multi-Image Causal Reasoning

Haoping Yu [1]    Yuanxi Li [1]    Jing Ma [1]

## Abstract

Visual causal reasoning is essential for understanding and intervening in the physical world, requiring identification of causal variables from visual inputs and reasoning over intervention effects. Despite recent progress, large vision–language models (VLMs) remain brittle at such tasks, especially for interventional and counterfactual queries over multi-image inputs. Most existing explorations inject causal knowledge via textual prompts, leaving causal mechanisms external to model execution and limiting reliable control during inference. To address this problem, we propose **BridgeVLM**, which internalizes visual causal reasoning by inducing a causal graph from multi-image inputs and converting it into structured **Causal Tokens** executed by **RAMP** layers injected into the LLM decoder for causal message passing. We further introduce a unified training interface **M3S** for fine-grained causal supervision from different granularities (local-/global level). BridgeVLM achieves **54.4%** accuracy on intervention tasks on CausalVLBench (vs. **33.2%** with prompt-level supervision), improves results on Causal3D from **43.6%** to **49.0%**, and substantially improves causal structure learning on CausalVLBench ($F_1$: **33.4%** $\rightarrow$ **75.1%**).

## 1. Introduction

Large Vision–Language Models (LVLMs) have emerged as powerful general-purpose assistants for multimodal instruction following and open-ended generation. However, as observed in previous work (Liu et al., 2025), they remain brittle on visual causal reasoning tasks, especially those involving *interventional target prediction* and *counterfactual*

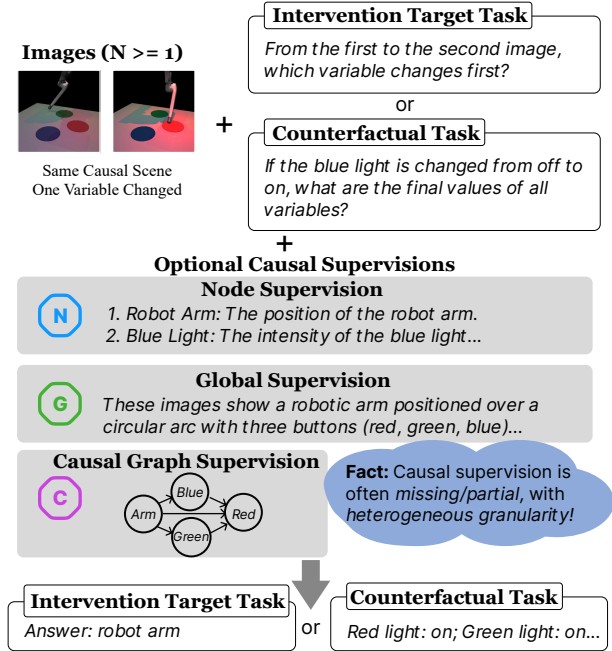

*Figure 1.* **Task and supervision overview**. Given image pairs/sequences from the same causal scene, the goal is to predict the manipulated variable or the resulting variable states. Optional causal supervision can aid prediction, but is typically imperfect.

queries (examples are shown in Figure 1). Interventional target prediction asks which variable was directly manipulated, given an original image and a post-intervention image in which intervening on one variable (e.g., a robot arm) may induce downstream changes in others (e.g., light colors). Counterfactual queries ask "what if" questions, such as how other variables would change if a variable (e.g., light color) were altered. Solving these tasks requires (i) *pinpointing the intervened variable* by comparing one or more images, and (ii) *propagating the intervention's effects* through the scene's causal relations to determine which other variables must change and how. This is challenging because current VLMs largely lack causal understanding, especially when images depict the same scene with only subtle, localized manipulations. (Komanduri et al., 2025; Liu et al., 2025).

A natural attempt is to inject causal mechanisms via **prompt-**

[1]Department of Computer & Data Sciences, Case Western University, Cleveland, OH, US. Correspondence to: Jing Ma <jxm1384@case.edu>.

*Proceedings of the 43rd International Conference on Machine Learning*, Seoul, South Korea. PMLR 306, 2026. Copyright 2026 by the author(s).

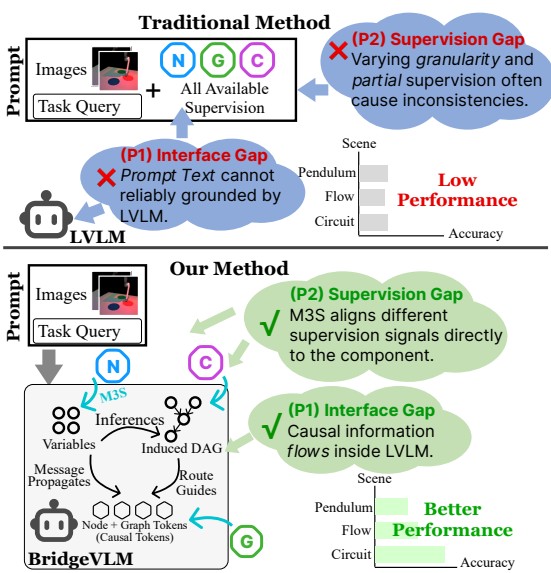

*Figure 2.* **Motivation of BridgeVLM.** Compared to traditional VLMs that concatenate images, queries, and supervision as prompts, BridgeVLM closes the *(P1) interface gap* by routing causal information inside the model and closes the *(P2) supervision gap* by aligning heterogeneous, imperfect supervision to the appropriate components via M3S, yielding better performance.

ing (Ma, 2025) (for example, appending text-form causal graphs, variable-relation descriptions, or explanation/rationale traces that finetuned by the model) to supervise the model (Figure 2, top). However, this straightforward strategy suffers from two obstacles. **(P1) Interface gap:** without an explicit *model-internal* interface, this causal supervision remains confined to *prompt text* level and cannot be reliably grounded in the internal representations that drive model decoding (Turpin et al., 2023; Paul et al., 2024). Especially, for LVLMs, language-dominant behavior further encourages reliance on textual priors over visual evidence, making long reasoning-style prompts an unreliable control substrate (Zhao et al., 2025; Vo et al., 2025). **(P2) Supervision gap:** in practice, causal supervision is often *missing or partial*, and frequently arrives at heterogeneous granularity (e.g., some data provide only global explanations, others include node/edge labels of causal graphs), since constructing high-quality causal graphs is also costly and error-prone (Komanduri et al., 2025; Chen et al., 2024; Liu et al., 2025). Our experimental results also substantiate these limitations: prompt-level causal supervision yields only marginal performance gains. For example, Phi-4-MMI-7B improves from 31.0% to 33.2% in intervention tasks on CausalVLBench and from 43.3% to 43.6% on Causal3D (Microsoft et al., 2025).

**Motivation.** Therefore, we argue that the bottleneck is not merely model scale or longer prompts, but the lack of an *internal interface* to supervise causal reasoning. Without an explicit structural representation aligned to intervenable variables, causal supervision (e.g., causal graphs, node/edge descriptions, structured traces) remains at the *prompt/reasoning output-text level*, which provides weak grounding and can even harm reasoning when prompts become long and distractive.

With this motivation, we propose **BridgeVLM**, a novel framework that resolves the two obstacles above by turning causal knowledge into *operational, internal* representations (Figure 2, bottom). To close the **interface gap (P1)**, BridgeVLM induces a *directed acyclic graph (DAG)* within the model, serving as a structural proxy for causal relationships among variables. The variables are captured in internal *variable features*, and then we *enforce* the information to flow among variables via the DAG with a **R**oute-**A**ware **M**essage **P**ropagation (RAMP), producing **Causal Tokens** that the decoder in LLM can attend to at every generation step. To close the **supervision gap (P2)**, we introduce **M3S** (**M**ulti-**S**ource **S**ignal **S**upervision), a unified interface that aligns heterogeneous supervision signals—including optional causal graph edges, node/edge/global descriptions, and structured traces—directly to the induced DAG and Causal Tokens. When ground-truth causal graphs are available, M3S can further refine the induced DAG toward the true causal graph.

We evaluate BridgeVLM on Causal3D and CausalVLBench benchmarks for interventional target prediction and counterfactual prediction (Liu et al., 2025; Komanduri et al., 2025). As shown in Table 1, BridgeVLM yields large gains over prompt-level mechanism supervision (e.g., 33.2% → 54.4% on CausalVLBench intervention; 84.8% → 90.0% on CausalVLBench counterfactual), and it also improves Causal3D counterfactual accuracy to 92.3% (vs. 81.0%). Notably, despite using a 7B backbone, BridgeVLM outperforms the strongest reported open-source LVLM baselines (up to 32B parameters) and is competitive with, and in these benchmarks slightly higher than, a strong closed-source commercial baseline (Komanduri et al., 2025; Google, 2024). Moreover, token-level supervision substantially improves causal graph recoverability on CausalVLBench (directed-edge F1: 33.4 → 75.1; Table 4), suggesting BridgeVLM learns a more faithful structural representation rather than merely a stronger end-task predictor.

**Contributions.** In this paper, we make the following pioneering contributions for multi-image interventional and counterfactual visual causal reasoning with LVLMs:

- **Internal interface for causal supervision.** To the best of our knowledge, this is the *first* work to bridge

the *interface gap* of causal supervision for LVLM in multi-image causal reasoning. To this aim, we induce a DAG from images and generate *Causal Tokens* for LLM decoder, while enforcing routing-aware message propagation (RAMP) along the DAG to support causal reasoning.

- **Unified supervision bridge for *missing* and *multi-granularity* causal signals.** We are the first to tackle the practical *supervision gap* (missing/partial supervision at heterogeneous granularity) for LVLM causal reasoning with **M3S**, to directly supervise the induced DAG and Causal Tokens using any available combination of causal graph and node/edge/global descriptions.

- **Experimental validation that internal causal supervision beats the prompt-level.** On Causal3D and CausalVLBench benchmarks, our extensive experiments show that supervising internal Causal Tokens (rather than feeding the same information as prompts) yields large gains on both intervention and counterfactual tasks and improves causal graph recoverability.

## 2. Related Work

**Benchmarks for interventional and counterfactual visual causal reasoning.** Recent benchmarks make interventional and counterfactual *visual* causal reasoning explicit and consistently show that LVLMs remain brittle in these settings. CELLO evaluates LVLMs with explicit causal graphs (Chen et al., 2024), while CausalVLBench and Causal3D provide targeted evaluation for intervention-target and counterfactual prediction from visual inputs (Komanduri et al., 2025; Liu et al., 2025). Text-only benchmarks such as CLadder and CausalBench further highlight that even strong LLMs struggle to reliably execute formal causal inference rules (Jin et al., 2023; Wang, 2024). These benchmarks motivate methods that improve *knowledge-grounded* causal reasoning from evidence, beyond prompt-only heuristics.

**Causal modeling inside vision–language systems.** A line of work introduces causal formalisms into vision–language pipelines, often for robustness or hallucination mitigation rather than intervention/counterfactual reasoning. For example, CDC analyzes CLIP adaptation through an SCM lens (Zhang et al., 2024), CLIP-ICM studies invariant causal knowledge for OOD robustness (Song et al., 2025), and CausalMM performs causal interventions over attention to mitigate modality-prior hallucinations (Zhou et al., 2025). Relatedly, causal-graphical modeling has also been used to guide knowledge-driven vision–language generation, but without exposing an intervenable variable-level interface for causal reasoning (Parascandolo et al., 2025). Overall, these approaches do not provide an LVLM with an *internal, tokenized, variable-level* causal-graph interface induced from

images and directly consumed during decoding.

**External causal knowledge: graph elicitation and prompting-level supervision.** Another thread obtains causal knowledge externally (e.g., querying LLMs for edges and reconciling them with causal constraints) (Long et al., 2024; 2023; Jiralerspong et al., 2024; Darvariu et al., 2024; Kampani et al., 2024; Wan et al., 2025), or injects causal information through prompts and explanation-style supervision (Zhang et al., 2025; Jiang et al., 2024). However, in both cases causal structure is either an *external artifact* or represented purely as *text*, leaving a gap between described causal knowledge and operational internal computation; moreover, generated rationales can be unfaithful to the actual decision process (Turpin et al., 2023; Paul et al., 2024). BridgeVLM addresses this gap by internalizing an induced *DAG* as a token-level interface and directly aligning heterogeneous supervision to the internal representation.

## 3. Method

In this section, we introduce our novel methods for multi-image visual causal reasoning, including BridgeVLM and M3S. BridgeVLM closes the *interface gap (P1)* by internalizing prompt-level causal knowledge as a *model-internal* interface. Specifically, it learns a routing DAG from images, with information flow constrained by the learned structure, and uses it to learn *Causal Tokens* that the decoder in LLM backbone can directly attend to. M3S closes the *supervision gap (P2)* by making the interface work even under *missing* and *multi-granularity* supervision, adaptively leveraging any available causal signals directly to supervise the learning of induced DAG and Causal Tokens. Figure 3 summarizes the architecture and training signals.

### 3.1. BridgeVLM: Enabling an Internal Interface for Causal Supervision

BridgeVLM augments an LVLM with a *model-internal interface* for causal supervision by introducing a set of internal causal components. These components can be learned from visual inputs even when explicit causal supervision is unavailable. Specifically, BridgeVLM (i) extracts latent variable features from images and induces a *routing DAG* over features, (ii) enforces DAG-based routing-aware message propagation (RAMP) to produce *Node Tokens*, (iii) Generate Graph Tokens, and update Node Tokens to produce *Causal Tokens*. These components enable a shift from fragile prompt-based supervision to *model-internal* supervision on representations that directly influence decoding.

Given inputs: images $\{\mathbf{I}_m\}_{m=1}^M$ (with $M \geq 1$) and a text query $x$; BridgeVLM generates *Causal Tokens* $\mathbf{T}^c$, and injecting them into the decoder in LLM for knowledge-grounded prediction. Let $H$ be the decoder hidden size.

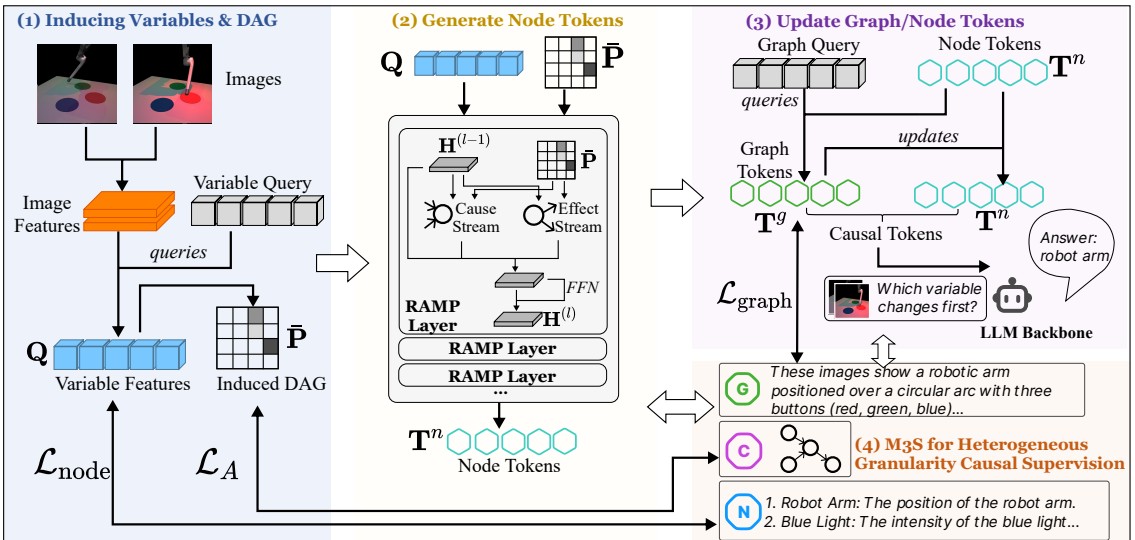

*Figure 3.* **Method overview. BridgeVLM** contains three stages: (1) Inducing latent variable features and DAG; (2) generate node tokens; and (3) generate and update causal tokens. **M3S** (4) further provides causal supervision at heterogeneous granularity.

We use $D$ as the maximum number of latent variable features and $G$ as the number of global Graph Tokens. Token sequences are represented as matrices in $\mathbb{R}^{(\cdot) \times H}$. $\mathrm{Attn}(\cdot)$ denotes standard multi-head attention/cross-attention.

### 3.1.1. MULTI-IMAGE ENCODING AND VARIABLE FEATURES

Each image $\mathbf{I}_m$ is encoded by a vision encoder $E_{\mathrm{vis}}$ into visual tokens $\mathbf{V}_m = E_{\mathrm{vis}}(\mathbf{I}_m) \in \mathbb{R}^{L_m \times H}$, where $L_m$ is the number of visual tokens for image $m$. We concatenate $\mathbf{V} = [\mathbf{V}_1; \ldots; \mathbf{V}_M] \in \mathbb{R}^{L \times H}$ with $L = \sum_{m=1}^{M} L_m$.

To obtain variable-level representations shared across images, we introduce learnable variable queries $\mathbf{Q}_0 \in \mathbb{R}^{D \times H}$ and extract variable features by cross-attending to the concatenated visual tokens:

$$\mathbf{Q} = \mathrm{Attn}(\mathbf{Q}_0, \mathbf{V}, \mathbf{V}) \in \mathbb{R}^{D \times H}. \quad (1)$$

Here $\mathbf{Q}$ serves as the initial *variable features* (one per potential intervenable variable). When a sample contains fewer than $D$ variables, we use a binary variable mask $\mathbf{m} \in \{0, 1\}^D$ to ignore invalid variables in losses and further operations.

**Intuition.** Variable feature extraction compresses visual evidence (single or multi-image) into a small set of variable-aligned representations, providing a clean interface for downstream structural reasoning and supervision.

### 3.1.2. INDUCING A DAG AS THE ROUTING BACKBONE

We induce a *routing DAG* over variables to serve as an explicit scaffold for *route-conditioned information flow*. The induced DAG is a kind of latent knowledge for internal reasoning (not a standalone causal graph recovery objective).

From variable features $\mathbf{Q}$, we predict directed adjacency logits with a low-rank parameterization:

$$\tilde{\mathbf{A}} = f_L(\mathbf{Q}) \, f_R(\mathbf{Q})^\top, \qquad \mathbf{P} = \sigma(\tilde{\mathbf{A}}) \in [0, 1]^{D \times D}, \quad (2)$$

where $f_L, f_R$ are MLPs producing $D \times r$ features (small rank $r$), and $\sigma(\cdot)$ is sigmoid. In practice, we (i) mask invalid variables using $\mathbf{m}$ and (ii) drop self-loops.

**Intuition.** $\mathbf{P}$ makes "who can influence whom" an explicit, differentiable object. It allows the model to condition computation on these relations and allows supervision (when available) to constrain the causal relations directly.

### 3.1.3. ROUTE-AWARE MESSAGE PROPAGATION (RAMP)

RAMP enforces *DAG-conditioned* propagation between variable features, producing route-consistent Node Tokens. Unlike global self-attention that mixes tokens indiscriminately, RAMP shapes information flow using the induced routing DAG, enabling variable-to-variable effect propagation required by interventions and counterfactuals.

We initialize node states with variable features $\mathbf{H}^{(0)} = \mathbf{Q} \in \mathbb{R}^{D \times H}$. For stability, we use a row-normalized matrix $\bar{\mathbf{P}} = \mathrm{RowNorm}(\mathbf{P})$ during propagation. Each RAMP layer

aggregates *effect stream* (along edges) and *cause stream* (reverse-direction) context with separate transforms:

$$\mathbf{M}^{(\downarrow)} = \bar{\mathbf{P}}\,\mathbf{H}^{(\ell-1)}\mathbf{W}_{\downarrow}^{(\ell)}, \tag{3}$$

$$\mathbf{M}^{(\uparrow)} = \bar{\mathbf{P}}^{\top}\mathbf{H}^{(\ell-1)}\mathbf{W}_{\uparrow}^{(\ell)}, \tag{4}$$

$$\mathbf{H}'^{(\ell)} = \mathrm{LN}\Big(\mathbf{H}^{(\ell-1)} + \mathbf{M}^{(\downarrow)} + \mathbf{M}^{(\uparrow)}\Big), \tag{5}$$

$$\mathbf{H}^{(\ell)} = \mathrm{LN}\Big(\mathbf{H}'^{(\ell)} + \mathrm{FFN}^{(\ell)}(\mathbf{H}'^{(\ell)})\Big), \tag{6}$$

where $\mathbf{W}_{\downarrow}^{(\ell)}, \mathbf{W}_{\uparrow}^{(\ell)} \in \mathbb{R}^{H \times H}$ are learnable parameters. After $L_p$ layers we obtain **Node Tokens** $\mathbf{T}^n = \mathbf{H}^{(L_p)} \in \mathbb{R}^{D \times H}$.

**Clarification.** Although the routing DAG is directed (cause $\rightarrow$ effect), inference can still benefit from bidirectional information flow when identifying intervention targets and answering counterfactual queries, since these tasks often require the model to infer causes from observed effects. RAMP therefore performs *direction-aware inference message passing*: it keeps the directionality explicit via separate transforms, while allowing information to flow in both directions to build better variable representations.

### 3.1.4. GRAPH TOKENS, CAUSAL TOKENS, AND DECODER INJECTION

Node Tokens encode variable-local states; Graph Tokens summarize global causal context. Together they form *Causal Tokens*, a compact token interface that the decoder can directly consult for knowledge-grounded generation.

We maintain $G$ learnable graph queries $\mathbf{G}_0 \in \mathbb{R}^{G \times H}$. Graph Tokens aggregate whole-graph context from Node Tokens and feed it back to nodes:

$$\mathbf{T}^g = \mathrm{Attn}(\mathbf{G}_0, \mathbf{T}^n, \mathbf{T}^n), \qquad \mathbf{T}^n \leftarrow \mathrm{Attn}(\mathbf{T}^n, \mathbf{T}^g, \mathbf{T}^g). \tag{7}$$

We define **Causal Tokens** as $\mathbf{T}^c = [\mathbf{T}^n; \mathbf{T}^g] \in \mathbb{R}^{(D+G) \times H}$.

Let $\mathbf{X} \in \mathbb{R}^{L_x \times H}$ be the embedding sequence of the input query $x$. We form the decoder input sequence by concatenation:

$$\mathbf{Z}_0 = [\mathbf{V}; \mathbf{T}^c; \mathbf{X}]. \tag{8}$$

The decoder then generates the prediction autoregressively.

**Intuition.** Causal reasoning about any node is rarely local: it depends on global graph context such as shared causes and effect pathways. We therefore implement a node $\rightarrow$ graph $\rightarrow$ node cycle, where Graph Tokens aggregate whole-graph information from Node Tokens and then feed it back to update each node. This yields node representations that are both variable-aligned and globally context-aware, improving knowledge-grounded inference for interventions

and counterfactuals. This node $\rightarrow$ graph $\rightarrow$ node design is related to inducing/global-token mechanisms in set and graph models (Lee et al., 2019; Cai et al., 2023), but here the global tokens summarize the induced routing DAG and feed global causal context back to variable-level tokens.

### 3.1.5. BASE AUTOREGRESSIVE OBJECTIVE

Since all models are finetuned for the downstream tasks, our base objective is standard teacher-forced autoregressive learning. Let $\mathbf{y}$ be the target output sequence (answer-only or an optional structured trace plus answer). We optimize:

$$\mathcal{L}_{\mathrm{LM}} = -\sum_{t=1}^{|\mathbf{y}|} \log p_\theta(y_t \mid \mathbf{y}_{<t}, \mathbf{I}_{1:M}, x). \tag{9}$$

**Optional structured trace.** When training with structured reasoning traces, we optionally apply a lightweight schema-conditioned attention routing; we omit details here and provide them in Appendix A.2–A.4.

## 3.2. M3S: Multi-Source Signal Supervision under Missing and Heterogeneous Labels

M3S closes the supervision gap (P2) by translating *any available* causal knowledge signals into direct supervision on the induced routing DAG and Causal Tokens. It supports missing labels and heterogeneous granularity (partial edges, node/edge/global descriptions), and can optionally refine the induced DAG toward a ground-truth causal graph when causal-graph supervision is provided.

M3S adds optional auxiliary losses on top of $\mathcal{L}_{\mathrm{LM}}$. Each loss is activated only when its corresponding supervision is available.

### 3.2.1. (OPTIONAL) CAUSAL-GRAPH EDGE SUPERVISION

If a ground-truth causal graph adjacency $\mathbf{A}^\star \in \{0, 1\}^{D \times D}$ is provided (possibly partially observed), we supervise the induced edge probabilities using Binary Cross-Entropy Loss(BCE):

$$\mathcal{L}_A = \mathrm{BCE}\big(\mathbf{P}, \mathbf{A}^\star\big), \tag{10}$$

that computed only on observed entries (and valid variables).

Importantly, edge supervision assumes that the induced variable features have a stable correspondence to the ground-truth node ordering. Therefore, graph-level supervision is most effective when variable identities are grounded by node-level alignment; otherwise, the oracle graph may be applied to mismatched latent variables.

### 3.2.2. (OPTIONAL) TOKEN SEMANTICS ALIGNMENT VIA DESCRIPTIONS

When node/edge/global descriptions are available, we align (i) variable features and Node Tokens to node descriptions ($\mathcal{L}_{\text{node}}$), and (ii) pooled Graph Tokens to graph-level descriptions ($\mathcal{L}_{\text{graph}}$) using a symmetric contrastive objective $\mathcal{L}_{\text{NCE}}$ (full formula in Appendix A.1). Because latent variable features are permutation-invariant, we first match predicted features to described node ids via Hungarian assignment (as in set prediction) and compute alignment losses after reordering (Carion et al., 2020; Kuhn, 1955).

**Causal DAG refinement.** If $\mathbf{A}^\star$ is not available, we encourage acyclicity using a NOTEARS-style regularizer $\mathcal{L}_{\text{dag}}$ to refine $\mathbf{P}$ toward a causal DAG with augmented-Lagrangian updates and schedule (Appendix A.5).

### 3.2.3. M3S OBJECTIVE

We define the M3S auxiliary loss as $\mathcal{L}_{\text{M3S}} = \lambda_A \, \mathbb{I}[\mathbf{A}^\star]\mathcal{L}_A + \lambda_{\text{dag}} \mathcal{L}_{\text{dag}} + \lambda_{\text{desc}} \, \mathbb{I}[\text{desc}]\mathcal{L}_{\text{desc}}$; where $\mathcal{L}_{\text{desc}} = \mathcal{L}_{\text{node}} + \mathcal{L}_{\text{graph}}$ denotes the description-alignment terms, $\lambda$ are scalar hyperparameters controlling the corresponding weights, and $\mathbb{I}[\cdot]$ is an indicator function that activates the corresponding term only when the required supervision is available.

The overall training objective is $\mathcal{L}_{\text{LM}} + \mathcal{L}_{\text{M3S}}$.

**Intuition.** M3S makes supervision *land* on the same internal interface the model uses for prediction: whenever partial causal knowledge signals are available, they directly shape the induced DAG and Causal Tokens rather than being relegated to prompt text.

## 4. Experiments

We organize our experiments around six questions: **(RQ1)** How does BridgeVLM compare to strong baselines (both SOTA and our backbone)? **(RQ2)** Which architectural components of BridgeVLM contribute most, and are generic structured tokens sufficient? **(RQ3)** Are *internal* causal knowledge signals more effective than *external* (prompt-level) signals? **(RQ4)** Within M3S, which supervision sources are most useful? **(RQ5)** How well does the induced routing DAG align with the ground-truth causal graph? **(RQ6)** Does BridgeVLM transfer/adapt to a different visual causal reasoning scenario? Each question is answered in Sections 4.3–4.7.

### 4.1. Benchmarks and Tasks

We evaluate BridgeVLM on two visual causal reasoning benchmarks, **Causal3D** (Liu et al., 2025) and **CausalVL-Bench** (Komanduri et al., 2025), covering both *intervention-target prediction* and *counterfactual prediction*. Causal3D

consists of synthetic 3D scenes with explicitly specified causal knowledge (i.e., rule-based structural relations), while CausalVLBench contains physically simulated scenarios (PENDULUM, FLOW, CIRCUIT). We follow the standard evaluation protocols and use an 8:1:1 train/validation/test split for all datasets (details in Appendix C). We report accuracy for downstream tasks and additionally report directed-edge F1 between the induced DAG and the ground-truth causal graph for CausalVLBench intervention (Appendix F).

### 4.2. Experimental Setup

**Backbone and variants.** We use **Phi-4-MMI-7B** as the backbone LVLM (Microsoft et al., 2025). **BridgeVLM** adds latent variable features, an induced *routing DAG*, RAMP, and **Causal Tokens**, trained with **M3S**. We report parameter, FLOP, and latency overhead in Appendix B.1.

**Baseline.** We use strong open-source LVLMs, including LLaVA-OneVision-7B (Li et al., 2024), DeepSeek-VL2-S-16B (Wu et al., 2024), and Qwen2.5-VL-32B (Bai et al., 2025). We also include the closed-source commercial VLM Gemini-2.0-Flash (Google, 2024). Some results are taken from the best results reported in the CausalVLBench paper (Komanduri et al., 2025). Note that their settings differs from our zero-shot evaluation, as it incorporates additional few-shot learning information that could enhance performance; we include these results alongside ours for reference.

**Fine-tuning protocol and supervision usage.** Unless marked *No Finetuning*, models under our backbone are jointly fine-tuned across all tasks/scenarios within each benchmark. We compare two ways of using the *same* causal supervision (global and per-node explanations; edge supervision when available):

- **Prompt-level (baseline):** append causal knowledge to the input prompt, optionally supervising generated explanations.

- **Internal-level (BridgeVLM):** use causal knowledge to directly supervise the internal Causal Tokens via M3S.

Unless explicitly stated, BridgeVLM does *not* use the decoding-time routing mechanism (Appendix A.2). All evaluations are performed without few-shot in-context exemplars.

### 4.3. Downstream Task Prediction: Main Results

**RQ1 (overall performance).** We compare BridgeVLM with (i) strong open-source and closed-source commercial

*Table 1.* **Main results on Causal3D (C3D) and CausalVLBench.** P/F/C denote PENDULUM/FLOW/CIRCUIT for CausalVLBench. For rows marked with "*", numbers are taken from CausalVLBench (Komanduri et al., 2025). **Bold** marks the best result including commercial (CM) models; underline marks the best result excluding CM models.

| | INTERVENTION-TARGET | | | | | COUNTERFACTUAL | | | | |
| | C3D | CAUSALVLBENCH | | | | C3D | CAUSALVLBENCH | | | |
| METHOD / SETTING | AVG | P | F | C | AVG | AVG | P | F | C | AVG |
|---|---|---|---|---|---|---|---|---|---|---|
| LLAVA-ONEVISION-7B (LI ET AL., 2024) | 39.8 | 27.1* | 32.7* | 35.9* | 31.9* | 60.1 | 83.5* | 85.0* | 96.9* | 88.5* |
| DEEPSEEK-VL2-S-16B (WU ET AL., 2024) | 33.8 | 24.4* | 34.4* | 28.1* | 29.0* | 53.0 | 77.9* | 51.4* | 41.6* | 57.0* |
| QWEN2.5-VL-32B (BAI ET AL., 2025) | – | 27.4* | 37.3* | 32.0* | 32.2* | – | 87.4* | 86.7* | 98.4* | **90.8*** |
| GEMINI-2.0-FLASH (CM) (GOOGLE, 2024) | 33.5 | **47.4*** | **55.7*** | 66.1* | **56.4*** | 65.0 | 86.5* | **88.3*** | 97.4* | 90.7* |
| PHI-4-MMI-7B (NO-FT, CAUSAL-PROMPT) | 38.7 | 26.4 | 23.7 | 26.3 | 25.4 | 77.6 | 63.0 | 73.3 | 94.0 | 76.8 |
| PHI-4-MMI-7B (ANSWER) | 43.3 | 31.2 | 22.1 | 39.8 | 31.0 | 81.6 | 76.2 | 80.9 | 95.6 | 84.2 |
| PHI-4-MMI-7B (CAUSAL-TRACE) | 43.6 | 29.2 | 26.2 | 44.3 | 33.2 | 81.0 | 76.5 | 83.1 | 94.9 | 84.8 |
| BRIDGEVLM-7B (OURS) | **49.0** | 36.0 | 43.6 | **83.7** | 54.4 | **92.3** | **87.7** | 83.7 | **98.5** | 90.0 |

LVLM baselines and (ii) controlled Phi-4 backbone variants under our fine-tuning protocol. Table 1 compares the performance of intervention target prediction and counterfactual prediction on both benchmarks. The Phi-4-MMI-7B baseline consists of three variants: (i) a non-finetuned model prompted with causal knowledge; (ii) a finetuned model trained directly using the answer as the objective; and (iii) a fine-tuned model trained with prompt-level causal-trace supervision. BridgeVLM-7B uses answer directly as the training objective, and causal knowledge as M3S supervision.

**Intervention:** BridgeVLM makes the *same* causal knowledge *substantially more useful* when applied at the internal Causal-Token/DAG interface: it improves over prompt-level causal injection ($33.2 \rightarrow 54.4$ Avg on CausalVLBench), with the largest gain on CIRCUIT ($44.3 \rightarrow 83.7$), consistent with scenarios requiring longer-range dependency propagation. BridgeVLM also surpasses the strongest reported open-source baselines (up to 32B) under the same benchmark, and is competitive with the commercial baseline (54.4 vs. 56.4 Avg; Table 1).

**Counterfactual:** BridgeVLM yields consistent improvements on both benchmarks (e.g., $84.8 \rightarrow 90.0$ Avg on CausalVLBench; $81.0 \rightarrow 92.3$ on Causal3D), achieving the best results on PENDULUM and CIRCUIT and remaining close to the strongest overall baseline (which has a 32B model size vs. 7B) on Avg.

### 4.4. Ablation: Which parts of BridgeVLM matter most?

**RQ2 (component importance).** We ablate one component at a time (Table 2) of Routing DAG (DAG), Node Token (NODE) and Graph Token (GRAPH), using the same training setup, to identify which parts drive intervention gains. All variants in Table 2 are evaluated with decoding-time routing enabled to maximize utilization of token be-

*Table 2.* **Component ablations on CausalVLBench intervention (accuracy %).** Numbers in parentheses are absolute drops relative to the full model (NONE).

| MASK | PENDULUM | FLOW | CIRCUIT | AVG |
|---|---|---|---|---|
| NONE | 39.6 | 53.4 | 86.4 | 59.8 |
| DAG | 22.7(-16.9) | 41.6(-11.8) | 67.9(-18.5) | 44.0(-15.8) |
| NODE | 34.6(-5.0) | 29.2(-24.2) | 78.6(-7.8) | 47.5(-12.3) |
| GRAPH | 25.0(-14.6) | 44.3(-9.1) | 69.6(-16.8) | 46.3(-13.5) |

haviors. (implementation details in Appendix A.2&D).

We further add a Slot-Attention baseline (Locatello et al., 2020) that preserves structured intermediate tokens but removes the routing DAG, RAMP, and Graph Tokens. It reaches 38.3 Avg on CausalVLBench intervention, above Phi-4 causal-trace (33.2) but far below BridgeVLM (54.4), indicating that generic slot tokens help but DAG-conditioned routing is the main source of the gain. The detailed result is illustrated in Appendix E. These results suggest that BridgeVLM's gain is not merely due to adding extra intermediate tokens. Rather, the benefit comes from coupling variable-level tokens with an induced routing DAG and global Graph Tokens, which jointly provide both local variable grounding and global causal context.

**Takeaway.** All three components matter, but removing the **routing DAG** consistently hurts performance across all scenarios ($59.8 \rightarrow 44.0$), confirming that directed routing is a primary source of gains. Graph Tokens are especially important on CIRCUIT and PENDULUM, where removing graph tokens causes a severe collapse in accuracy, consistent with the need for long-range/global dependency aggregation. Node Tokens are most critical on FLOW, where removing node-level tokens leads to the largest drop ($53.4 \rightarrow 29.2$), suggesting that fine-grained variable representations are necessary for node-sensitive intervention reasoning.

*Table 3.* **Supervision ablation on CausalVLBench intervention (accuracy %).** **Answer**: Base model, trained directly using the answer. **GraphPrompt**: ground-truth causal graph as prompt. **NodePrompt**: node/global descriptions as prompt. **GraphAlign**: supervise DAG with ground-truth causal graph adjacency (M3S causal graph supervision) . **NodeAlign**: align Causal Tokens to node/global descriptions (M3S token alignment). **AttBias**: Decoding time attention routing is applied.

| Variant (CausalVLBench Intervention) | Pendulum | Flow | Circuit | Avg |
|---|---|---|---|---|
| Phi-4 (FT, Answer) | 31.2 | 22.1 | 39.8 | 31.0 |
| Bridge (Answer) | 32.9 | 34.4 | 39.4 | 35.6 |
| *External (prompt-level) signals for RQ3* | | | | |
| Bridge (+GraphPrompt) | 25.4 | 44.6 | 38.3 | 36.1 |
| Bridge (+NodePrompt) | 24.5 | 22.0 | 38.3 | 28.3 |
| Bridge (+GraphPrompt+NodePrompt) | 25.4 | 22.0 | 38.3 | 28.5 |
| *Internal (M3S) signals for RQ3/RQ4* | | | | |
| Bridge (+GraphAlign) | 28.2 | 28.7 | 36.1 | 31.0 |
| Bridge (+NodeAlign) | 36.0 | 43.6 | 83.7 | 54.4 |
| Bridge (+NodeAlign+AttBias) | **39.6** | 53.4 | **86.4** | 59.8 |
| Bridge (+NodeAlign+AttBias+GraphAlign) | 39.3 | **55.0** | 86.4 | **60.2** |

## 4.5. Ablation: Which supervision signals matter?

**RQ3 (external vs. internal) & RQ4 (which M3S signals).** We compare prompt-level causal injection with internal-level supervision on the same causal knowledge, then ablate which internal supervision sources contribute most. Table 3 highlights a clear separation between *external* (prompt-level) and *internal* level supervision; and toggles M3S supervision channels while keeping BridgeVLM's architecture fixed, to isolate the contribution of each supervision channel.

**Takeaway.** The result indicates causal graph or textual descriptions as *prompts* (**GraphPrompt/NodePrompt**) provides little gain and can even be unstable relative to answer-only training compare to *internal alignments*, indicating that prompt-level causal information is not reliably utilized (**RQ3**). In contrast, **NodeAlign** (aligning internal Causal Tokens) is the main driver of gains, showing that causal supervision becomes effective when it directly shapes the model's internal variable-level representations (**RQ4**).

Also, **GraphAlign** (supervising the routing DAG with the ground-truth causal graph) alone yields no improvement, or even worse for some tasks. We infer that it is caused by node-graph misalignment: without grounded node semantics (via **NodeAlign**), edge supervision provides a graph whose endpoints are not consistently tied to the variable features, so this supervision itself cannot induce the intended information flow. When combined with **NodeAlign**, **GraphAlign** only brings a marginal gain, suggesting that BridgeVLM can benefit from learning a *reasonably plausible* routing DAG; once node tokens are semantically grounded, enforcing an exactly matched causal graph offers diminishing returns for the downstream task.

A post-hoc slot-node matching diagnostic in Appendix F.1 further supports this explanation: without NodeAlign, variable-to-node correspondence remains unstable even when GraphAlign is applied.

This does not imply that causal prompts are universally harmful; rather, in our multi-image setting, long structured prompts appear difficult for the LVLM to ground consistently in visual evidence. Internal supervision avoids this issue by applying the same causal knowledge directly to the representations that participate in decoding.

Note that **AttBias** yields only a very small extra boost and is treated as an optional trick; therefore, we do not include it in the default BridgeVLM results except Section 4.4.

## 4.6. Graph Recovery on CausalVLBench: Does BridgeVLM learn a more faithful structure?

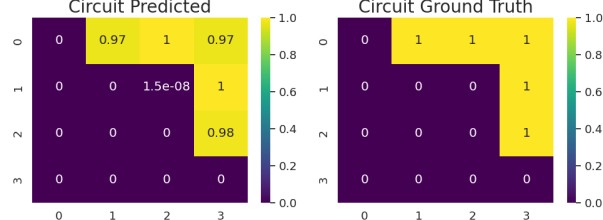

*Figure 4.* Visualization examples of induced DAG for Circuit scenario on CausalVLBench.

**RQ5 (structure diagnostic).** We use causal graph recovery on CausalVLBench interventions as a diagnostic for whether internal supervision – none (Ans-only), node explanations (NodeAlign) or supervising the routing DAG with the ground-truth causal graph as oracle adjacency (GraphAlign) – makes the induced routing DAG more aligned with the ground-truth causal graph. We report directed-edge F1 after binarization (Table 4 and Appendix F). Figure 4 shows an example induced DAG.

*Table 4.* **Graph recovery on CausalVLBench intervention.** We report F1 score for directed edge recovery by comparing the induced DAG with the ground-truth causal graph after binarization.

| GRAPH RECOVERY (DIRECTED-EDGE F1, %) | OVERALL |
|---|---|
| BRIDGEVLM (ANS-ONLY) | 33.4 |
| BRIDGEVLM (+GRAPHALIGN) | 100.0 |
| BRIDGEVLM (+NODEALIGN) | 75.1 |

*Table 5.* **CELLO intervention accuracy (%).** BridgeVLM does not claim zero-shot OOD generalization, but improves over the adapted backbone after CELLO adaptation. **CELLO**: Already fine-tuned for CELLO dataset. **CVLB**: Finetuned on CausalVLBench dataset only.

| METHOD | CoI | BAS | CDE | AVG |
|---|---|---|---|---|
| PHI-4 (NO-FT, 0-SHOT) | 60.0 | 60.0 | 78.5 | 66.2 |
| PHI-4 (CELLO) | 80.0 | 76.0 | 95.2 | 83.7 |
| BRIDGE (CVLB, 0-SHOT) | 58.6 | 48.6 | 82.2 | 63.1 |
| BRIDGE (CELLO) | **94.3** | **80.0** | **96.6** | **90.3** |

**Takeaway.** Although the model is trained jointly across multiple scenarios, the oracle adjacency head reaches 100% by construction, indicating the model can reliably distinguish different causal graphs across scenarios. Token-level description alignment substantially improves graph recovery, and higher recovery is consistently associated with stronger intervention accuracy, suggesting that better structural alignment supports downstream interventional reasoning.

### 4.7. Can BridgeVLM handle out-of-distribution and single-image reasoning scenarios?

**RQ6 (transfer/adaptation).** Although our work primarily targets the already challenging setting of *multi-image visual causal reasoning within a fixed scene family*, we further evaluate whether BridgeVLM remains useful beyond this main in-distribution setup. Specifically, we test on the CELLO intervention task (Chen et al., 2024), a single-image causal reasoning benchmark with explicit causal graphs. As shown in Table 5, BridgeVLM trained only on CausalVL-Bench does not solve zero-shot cross-benchmark transfer (63.1% Avg vs. 66.2% for Phi-4-MMI No-FT), indicating that unseen out-of-distribution causal scenarios remain challenging. However, after CELLO fine-tuning, BridgeVLM reaches 90.3% average accuracy, outperforming the adapted Phi-4 baseline (83.7%). This suggests that the Causal-Token interface remains beneficial when adapted to a new causal benchmark, even in a single-image setting that emphasizes scene-level causal understanding rather than causal relations inferred from multi-image differences.

## 5. Conclusion

Our paper argues that a central obstacle for visual causal reasoning in LVLMs is the absence of a *model-internal* interface, as causal supervision purely from a prompt is difficult to take effect. We propose **BridgeVLM**, which induces a *DAG* from (single- or) multi-image inputs and internalizes it as **Causal Tokens** for decoding, with **RAMP** enabling DAG-conditioned message propagation over latent variables. With **M3S**, BridgeVLM can leverage missing, partial, and heterogeneous supervision signals to improve both end-task performance and the recoverability of the induced DAG, and can optionally refine this graph toward a causal DAG when causal graph supervision is available. Empirically, BridgeVLM achieves strong results with a comparatively small 7B backbone, surpassing larger open-source LVLMs and remaining competitive with a strong closed-source commercial model under the same benchmark protocols. Overall, our findings highlight that *where* supervision is applied (prompt- vs. internal-level) is critical for translating causal information into grounded reasoning.

**Limitations.** The induced DAG is a latent structure designed for internal reasoning rather than a guaranteed recovery of the true underlying causal graph without additional assumptions. In addition, our evaluation primarily assumes that test instances come from scenarios seen during training; generalization to entirely unseen causal scenarios (e.g., new variable sets or causal knowledge) remains a direction for future work.

## Impact Statement

This work aims to bridge vision–language modeling and causal learning by introducing a model-internal causal supervision interface for visual causal reasoning. By moving beyond prompt-level supervision and internalizing causal structure into learnable representations, our approach opens new opportunities for integrating causal reasoning directly into large multimodal foundation models. We hope this work encourages broader collaboration between the vision, causal inference, and foundation model communities, fostering shared benchmarks, interfaces, and learning principles for causal supervision in high-capacity models. Such cross-disciplinary efforts are essential for advancing models that can reason reliably about interventions, counterfactuals, and structured word dynamics rather than relying on surface correlations. More broadly, this direction holds promise for real-world applications where causal reasoning is critical, including robotics, embodied AI, scientific discovery, healthcare, and decision-support systems. By enabling foundation models to better represent and reason about causal structure, this work contributes toward more interpretable, robust, and trustworthy multimodal AI systems, and highlights several open challenges, such as generalization to unseen causal scenarios, that motivate future research. Our work does not raise any ethical concerns that need disclosure.

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

# A. Additional Implementation Details

## A.1. Symmetric InfoNCE for Node/Graph Description Alignment

We use a symmetric InfoNCE objective to align visual-derived representations (variable features / Node Tokens / Graph Tokens) with text-derived description embeddings.

When textual descriptions exist, we align (i) variable features $\mathbf{Q}$ and (ii) Node Tokens $\mathbf{T}^n$ to node-description embeddings, and align pooled Graph Tokens to graph-level description embeddings using a symmetric contrastive objective to get $\mathcal{L}_{\text{desc}}$:

$$\mathcal{L}_{\text{node}} = \beta_q \, \mathcal{L}_{\text{NCE}}(\tilde{\mathbf{Q}}, \mathbf{U}^n) + \beta_n \, \mathcal{L}_{\text{NCE}}(\tilde{\mathbf{T}}^n, \mathbf{U}^n), \tag{11}$$

$$\mathcal{L}_{\text{graph}} = \mathcal{L}_{\text{NCE}}(\text{MeanPool}(\mathbf{T}^g), \text{MeanPool}(\mathbf{U}^g)), \tag{12}$$

where $\mathbf{U}^n$ stacks node-description embeddings and $\mathbf{U}^g$ is the stacks of global and edge description embeddings. Here $\tilde{\mathbf{Q}}$ and $\tilde{\mathbf{T}}^n$ denote variable features and Node Tokens permuted to match the ground-truth node order.

**Symmetric InfoNCE (node-level, within-sample negatives).** For a sample $b$ with $n_b$ valid nodes, let $\mathbf{g}_i^{(b)}$ denote the visual-side representation for node $i$ (either $\mathbf{q}_i^{(b)}$ or $\mathbf{t}_i^{n,(b)}$), and let $\mathbf{u}_i^{(b)}$ denote the corresponding node-description embedding. We apply learned projections and $\ell_2$ normalization:

$$\hat{\mathbf{g}}_i^{(b)} = \text{norm}\big(f_g(\mathbf{g}_i^{(b)})\big), \qquad \hat{\mathbf{u}}_i^{(b)} = \text{norm}\big(f_u(\mathbf{u}_i^{(b)})\big). \tag{13}$$

We define pairwise similarities with temperature $\tau$:

$$s_{ij}^{(b)} = \frac{\hat{\mathbf{g}}_i^{(b)\top} \hat{\mathbf{u}}_j^{(b)}}{\tau}. \tag{14}$$

The symmetric InfoNCE loss for sample $b$ is:

$$\mathcal{L}_{\text{NCE}}^{(b)}(\mathbf{g}, \mathbf{u}) = \frac{1}{2}\Big( -\frac{1}{n_b} \sum_{i=1}^{n_b} \log \frac{\exp(s_{ii}^{(b)})}{\sum_{j=1}^{n_b} \exp(s_{ij}^{(b)})} - \frac{1}{n_b} \sum_{i=1}^{n_b} \log \frac{\exp(s_{ii}^{(b)})}{\sum_{j=1}^{n_b} \exp(s_{ji}^{(b)})} \Big). \tag{15}$$

We average $\mathcal{L}_{\text{NCE}}^{(b)}$ over batch samples that provide node descriptions, and apply the same form to graph-level alignment by treating each sample as one "instance" (with negatives from other batch samples).

**Symmetric InfoNCE (graph-level, across-batch negatives).** Let $\bar{\mathbf{t}}^{g,(b)}$ be a pooled representation of Graph Tokens for sample $b$ (e.g., mean pooling over $G$ tokens), and let $\mathbf{u}^{g,(b)}$ be the graph-level description embedding. We treat each sample as one instance and use other batch elements as negatives via a symmetric contrastive objective (equivalently, symmetric cross-entropy on the $B \times B$ similarity matrix).

**Hungarian bipartite matching for variable identity alignment.** Let $\mathbf{T}^{pred,(b)} \in \mathbb{R}^{D_p \times H}$ be predicted node representations used for matching (we use variable features $\mathbf{Q}$ by default, and fall back to Node Tokens when needed), and let $\mathbf{U}^{(b)} \in \mathbb{R}^{D_g \times H}$ be node-description embeddings in a fixed ground-truth order. Let $\mathbf{m}^{(b)} \in \{0,1\}^{D_g}$ denote the valid-node mask and define the valid index set $\mathcal{J}^{(b)} = \{j \mid \mathbf{m}_j^{(b)} = 1\}$.

We compute a similarity matrix:

$$S_{i,j}^{(b)} = \text{norm}(\mathbf{T}_i^{pred,(b)})^\top \text{norm}(\mathbf{U}_j^{(b)}), \qquad i \in \{1, \ldots, D_p\}, \; j \in \mathcal{J}^{(b)}. \tag{16}$$

We then solve a one-to-one assignment from valid described nodes to predicted variables:

$$\pi^{(b)} = \arg\max_{\pi} \sum_{j \in \mathcal{J}^{(b)}} S_{\pi(j),j}^{(b)} \quad \text{s.t. } \pi(j) \in \{1, \ldots, D_p\} \text{ and } \pi(j) \neq \pi(j') \, \forall j \neq j'. \tag{17}$$

Equivalently, we minimize the cost $C_{i,j}^{(b)} = -S_{i,j}^{(b)}$ and solve it with the Hungarian algorithm (Kuhn, 1955) (as commonly done for set prediction in DETR (Carion et al., 2020)).

**Permutation and aligned losses.** After obtaining $\pi^{(b)}$, we reorder predicted node tokens to match the ground-truth node order:

$$\tilde{\mathbf{T}}_j^{n,(b)} = \mathbf{T}_{\pi^{(b)}(j)}^{n,(b)}, \qquad \forall j \in \mathcal{J}^{(b)}. \tag{18}$$

If we also supervise the latent DAG, we apply the same permutation to both rows and columns:

$$\tilde{\mathbf{A}}_{j,k}^{(b)} = \mathbf{A}_{\pi^{(b)}(j),\,\pi^{(b)}(k)}^{(b)}, \qquad \forall j,k \in \mathcal{J}^{(b)}. \tag{19}$$

All node-/graph-alignment and DAG supervision losses are then computed on these aligned (permuted) predictions, with masks applied to ignore invalid nodes/edges.

**Implementation notes.** (i) We apply the node-presence mask so that only valid nodes contribute. (ii) We optionally use cross-batch negatives via a memory queue; in our default implementation we use in-batch negatives for simplicity and reproducibility.

## A.2. Optional Schema-Conditioned Decoding-Time Routing for Structured Reasoning

**Schema-conditioned routing is an optional decoding-time mechanism for *structured* reasoning traces. It biases attention based on the current schema field (e.g., node-specific, edge-specific, global explanation, answer), so each field preferentially consults the most relevant subset of Causal Tokens—improving precision without changing the base model architecture.**

### A.2.1. SCHEMA STATE AND STRUCTURED OUTPUTS

When BridgeVLM is asked to produce a reasoning trace, it uses a simple schema with sentinel tokens (or function-calling style arguments) to delimit fields. A lightweight state machine parses the generated token and outputs a schema state $\phi_t$ at decoding step $t$:

$$\phi_t \in \{\mathsf{Node}(k), \mathsf{Edge}(k,\ell), \mathsf{Explain}, \mathsf{Answer}, \mathsf{Other}\}. \tag{20}$$

The concrete sentinel design and parsing rules are provided in Appendix A.3.

### A.2.2. ATTENTION WITH ROUTING BIAS

For a transformer layer, the pre-softmax attention logit from query position $t$ to key position $j$ is:

$$a_{t,j} = \frac{\mathbf{q}_t^\top \mathbf{k}_j}{\sqrt{H}} + \mathbf{M}_{t,j} + \eta\,\mathbf{F}_{t,j}, \tag{21}$$

where $\mathbf{M}_{t,j}$ is the standard causal mask, $\eta$ is a scalar routing strength, and $\mathbf{F}_{t,j}$ is a schema-conditioned routing bias. Intuitively, $\mathbf{F}_{t,j}$ upweights the Causal Tokens relevant to the current schema state (e.g., the Node Token for $\mathsf{Node}(k)$) and optionally downweights irrelevant Node Tokens. The precise definition of $\mathbf{F}_{t,j}$ via focus/downweight sets is given in Appendix A.4.

**Practical note.** When routing is disabled, we set $\eta = 0$, recovering standard decoding.

## A.3. Schema State Machine and Sentinel Design

We recommend using explicit sentinel tokens to delimit structured fields. One concrete Phi-4-instruct style format is:

```
<|Node|> (k) ...
<|Edge|> (k -> l) ...
<|Explain|> ...
<|Assistant|> Answer: ...
```

A lightweight state machine scans the generated token and outputs $\phi_t$:

- If the latest special token is `<|Node|> (k)`, set $\phi_t = \mathsf{Node}(k)$ until next special token.

*Table 6.* Hyperparameter for reproducibility.

| CATEGORY | HYPERPARAMETERS |
|---|---|
| VISION ENCODER | $E_{\mathrm{vis}}$: FROM PHI-4-MM-INSTRUCT, FINETUNED FOR LAST 4 LAYERS. |
| VARIABLE FEATURES | $D$: 10 |
| GRAPH INDUCTION | $r$: 10 |
| RAMP | $L_p$: 4, DROPOUT: 0.1 |
| GRAPH TOKENS | $\mathbf{T}^n$: 10, $\mathbf{T}^g$: 10 |
| LOSS WEIGHTS | $\lambda_{\mathrm{TEXT}} : 0.5, \lambda_A : 0.5, \lambda_{\mathrm{NODE}} : 0.2, \lambda_{\mathrm{GRAPH}} : 0.2, \lambda_{\mathrm{DAG}}: 0.2$ |
| OPTIMIZATION | LEARNING RATE: 1E-4, BATCH SIZE: 8, WEIGHT DECAY: 0.01 |
| TRAINING | EPOCHS: 5, EARLY STOPPING: YES |

- If the latest special token is `<|Edge|>` (k -> l), set $\phi_t = \mathsf{Edge}(k, \ell)$ until next special token.

- If the latest special token is `<S_EXPLAIN>`, set $\phi_t = \mathsf{Explain}$.

- If the latest special token is `<|Assistant|> Answer`, set $\phi_t = \mathsf{Answer}$.

- Otherwise, $\phi_t = \mathsf{Other}$.

**Design recommendations.** (i) Use special tokens from tokenizer that are unique and unlikely to appear in natural text. (ii) Always include explicit node ids / edge endpoints to enable id-aware routing. (iii) For safety, if malformed patterns occur, fall back to $\mathsf{Other}$.

### A.4. Precise Routing Bias Definition via Focus/Downweight Sets

Let $\mathcal{I}_k^n$ be the position index of the injected Node Token for node $k$, and let $\mathcal{I}^g$ be the set of injected Graph Token indices (size $G$). Define $\mathcal{I}^n = \cup_{k=1}^D \mathcal{I}_k^n$ and $\mathcal{I}^c = \mathcal{I}^n \cup \mathcal{I}^g$. We define a focus set $\mathcal{F}(\phi_t)$ and a downweight set $\mathcal{D}(\phi_t)$:

$$\mathcal{F}(\mathsf{Node}(k)) = \mathcal{I}_k^n, \qquad \mathcal{D}(\mathsf{Node}(k)) = \mathcal{I}^n \setminus \mathcal{I}_k^n, \tag{22}$$

$$\mathcal{F}(\mathsf{Edge}(k,\ell)) = \mathcal{I}^g \cup \mathcal{I}_k^n \cup \mathcal{I}_\ell^n, \qquad \mathcal{D}(\mathsf{Edge}(k,\ell)) = \mathcal{I}^n \setminus (\mathcal{I}_k^n \cup \mathcal{I}_\ell^n), \tag{23}$$

$$\mathcal{F}(\mathsf{Explain}) = \mathcal{I}^c, \qquad \mathcal{D}(\mathsf{Explain}) = \emptyset, \tag{24}$$

$$\mathcal{F}(\mathsf{Answer}) = \emptyset, \qquad \mathcal{D}(\mathsf{Answer}) = \emptyset, \tag{25}$$

$$\mathcal{F}(\mathsf{Other}) = \emptyset, \qquad \mathcal{D}(\mathsf{Other}) = \emptyset. \tag{26}$$

Then the routing bias is:

$$\mathbf{F}_{t,j} = \mathbb{I}[j \in \mathcal{F}(\phi_t)] - \gamma \, \mathbb{I}[j \in \mathcal{D}(\phi_t)], \tag{27}$$

where $\gamma > 0$ controls how strongly we downweight irrelevant Node Tokens for node-/edge-specific fields.

### A.5. Augmented Lagrangian Optimization for NOTEARS-style Acyclicity

We use the NOTEARS acyclicity measure $h(\mathbf{P}) = \mathrm{tr}(\exp(\mathbf{P} \odot \mathbf{P})) - D$ and optimize an augmented Lagrangian:

$$\mathcal{L}_{\mathrm{dag}} = \alpha \, h(\mathbf{P}) + \frac{\rho}{2} h(\mathbf{P})^2. \tag{28}$$

We update the Lagrange multiplier $\alpha$ and penalty $\rho$ as:

$$\alpha \leftarrow \alpha + \rho \, h(\mathbf{P}), \qquad \rho \leftarrow \min(\rho_{\max}, c_\rho \rho), \tag{29}$$

where $c_\rho > 1$. In our default schedule, we (i) warm up without $\mathcal{L}_{\mathrm{dag}}$ for $T_{\mathrm{warm}}$ steps to stabilize variable/graph induction, then (ii) enable $\mathcal{L}_{\mathrm{dag}}$ and update $(\alpha, \rho)$ every $T_{\mathrm{update}}$ steps. We clip gradients on $\tilde{\mathbf{A}}$ for stability and optionally stop increasing $\rho$ once $h(\mathbf{P})$ plateaus.

## B. Hyperparameter Table

Table 6 provides a default hyperparameters we used for training & finetuning.

*Table 7.* **Efficiency comparison.** Latencies are measured per batch under our implementation.

|  | PHI-4-MMI-7B | BRIDGEVLM |
|---|---|---|
| TOTAL PARAMS | 5.574B | 6.359B |
| BACKBONE PARAMS (INCL. LORA) | – | 5.714B |
| VARIABLE FEATURE ENCODER | – | 226.68M |
| RAMP MODULE | – | 188.80M |
| CAUSAL TOKEN ENCODER | – | 230.15M |
| FLOPS / FORWARD | 11.713T | 13.009T |
| INFERENCE LATENCY / BATCH | 128.26 MS | 228.97 MS |
| TRAINING LATENCY / BATCH | 186.22 MS | 288.54 MS |

### B.1. Efficiency Analysis

We measure efficiency under the same setting as Appendix B using 2 NVIDIA A100 GPUs. Table 7 reports parameter count, per-forward FLOPs, and latency.

BridgeVLM introduces a moderate efficiency–performance tradeoff: it adds causal modules on top of the same Phi-4 backbone, resulting in additional parameters and latency but substantial gains in visual causal reasoning. The current latency likely overestimates the intrinsic overhead, since the Phi-4 backbone benefits from optimized kernels whereas our causal branch is implemented with standard PyTorch operators and is not yet kernel-optimized.

## C. Dataset Preparation

Dataset statistics is shown in table 8.

### C.1. Causal3D

For each scenario in Causal3D, we construct 3,000 samples for intervention target prediction, with balanced labels (1,000 samples per label for the 3 candidate variables); and 1,000 samples for counterfactual prediction, with balanced labels (250 samples per variable change for the 4 candidate variables) We use a stratified 8:1:1 train/val/test split *within each label* to ensure label balance across splits. That is, 9,000 samples in total.

For all Causal3D tasks, each example includes 3 input images. In *intervention-target prediction*, two images are non-intervened references and the third image reflects the intervention; in *counterfactual prediction*, the three images correspond to the scenario-specific counterfactual setup.

### C.2. CausalVLBench

We use the original CausalVLBench data, consisting of 17,744 PENDULUM samples, 10,890 FLOW samples, and 2,871 CIRCUIT samples. We apply the same stratified 8:1:1 split *within each label* for each scenario.

For CausalVLBench, *intervention-target prediction* task contains 2 images, where one is non-intervened and the other is intervened; *counterfactual prediction* task contains only 1 image, which corresponding to the scenario-specific setup.

### C.3. Overall Metric Under Scenario Imbalance

CausalVLBench scenarios are highly imbalanced for each scenario. Therefore, when reporting **average** accuracy in the main tables, we compute an unweighted mean of per-scenario accuracies (macro-average over PENDULUM/FLOW/CIRCUIT), rather than computing total correct predictions divided by total number of samples.

## D. Component Ablations of BridgeVLM

We evaluate the importance of BridgeVLM components by ablating one component at a time and re-evaluating intervention accuracy on CausalVLBench (Table 2). All ablations keep the remaining architecture and training setup unchanged; only the specified component is removed or masked.

*Table 8.* Dataset sizes and stratified 8:1:1 train/val/test splits. Splits are stratified by label within each scenario.

| DATASET | SCENARIO | #LABELS | TOTAL | TRAIN | VAL | TEST |
|---------|----------|---------|-------|-------|-----|------|
| CAUSAL3D-INTERVENTION | (PER SCENARIO) | 3 | 3,000 | 2,400 | 300 | 300 |
| CAUSAL3D-INTERVENTION | ALL (3 SCENARIOS) | 3 | 9,000 | 7,200 | 900 | 900 |
| CAUSAL3D-COUNTERFACTUAL | (PERSCENARIO) | 4 | 1,000 | 800 | 100 | 100 |
| CAUSAL3D-COUNTERFACTUAL | ALL (3 SCENARIOS) | 4 | 3,000 | 2,400 | 300 | 300 |
| CAUSALVLBENCH | PENDULUM | 4 | 17,744 | 14,196 | 1,774 | 1,774 |
| CAUSALVLBENCH | FLOW | 4 | 10,890 | 8,712 | 1,089 | 1,089 |
| CAUSALVLBENCH | CIRCUIT | 4 | 2,871 | 2,297 | 287 | 287 |
| CAUSALVLBENCH | ALL (3 SCENARIOS) | 4 | 31,505 | 25,205 | 3,150 | 3,150 |

*Table 9.* **Structured-token baseline on CausalVLBench intervention.** Slot-Attention preserves structured intermediate representations but removes the routing DAG, RAMP, and Graph Tokens.

| METHOD | PENDULUM | FLOW | CIRCUIT | AVG |
|--------|----------|------|---------|-----|
| PHI-4-MMI-7B (CAUSAL-TRACE) | 29.2 | 26.2 | 44.3 | 33.2 |
| SLOT-ATTENTION BASELINE | 34.5 | 43.3 | 37.0 | 38.3 |
| BRIDGEVLM-7B | **36.0** | **43.6** | **83.7** | **54.4** |

**Ablation definitions.**

1. **DAG** To remove the effect of the induced DAG structure, we replace the learned DAG with an all-ones adjacency before normalization, yielding a fully connected and uniform routing pattern for message passing (i.e., the model can no longer exploit directed structure).

2. **Node (remove RAMP and node-token injection).** To test the contribution of node-level structural tokens and RAMP propagation, we bypass RAMP and do not inject Node Tokens into the decoder. We still compute Graph Tokens from the variable features to preserve a global summary pathway.

3. **Graph (remove global tokens and local–global fusion).** To test the contribution of global structural context, we mask Graph Tokens and disable the graph-to-node update, so the decoder receives only Node Tokens without global fusion.

# E. Slot-Attention Baseline Comparison

To determine whether the gains come merely from using structured intermediate representations or specifically from our DAG-conditioned routing design, we introduce a closer structured baseline based on Slot Attention. This baseline replaces BridgeVLM's latent variable queries, routing DAG, RAMP layers, and Graph Tokens with Slot Attention, and injects the resulting slot tokens directly into the decoder. Thus, it preserves structured intermediate tokens while removing causal routing. As shown in Table 9, this baseline improves over the Phi-4 backbone with causal-trace supervision (33.2 Avg), but remains far below BridgeVLM (54.4 Avg). This indicates that structured tokens are helpful, but insufficient to explain the full gain. DAG-conditioned routing is a key contributor: it not only introduces structure, but also organizes causal information into Node Tokens for variable-specific reasoning and Graph Tokens for global causal context.

# F. Causal Graph Recovery on CausalVLBench

We evaluate graph recovery on CausalVLBench intervention by comparing the induced DAG $\mathbf{P} \in [0, 1]^{D \times D}$ with the ground-truth causal graph $\mathbf{A}^{\star} \in \{0, 1\}^{D \times D}$. We binarize predictions with a fixed threshold of **0.5** (i.e., predict an edge if $P_{ij} \geq 0.5$). We then compute directed-edge **precision/recall/F1** and accuracy from the aggregated $\mathrm{TP/FP/FN/TN}$ counts. We **ignore the diagonal** (self-edges are excluded). All confusion counts are **micro-aggregated** over the entire test set and reported as a single **overall** score across the three scenarios (PENDULUM, FLOW, CIRCUIT).

Table 10 shows that graph recovery quality is positively associated with downstream intervention performance across scenarios: higher recovery F1 tends to coincide with higher intervention and counterfactual accuracy.

Figure 5 provides visualization examples of induced DAG for the BridgeVLM with token-alignment supervision variant. The predicted edges are mostly consistent with the ground-truth structure.

*Table 10.* Causal Graph recovery (directed-edge F1) versus downstream task performance on CausalVLBench.

| SCENARIO | RECOVERY F1(%) | INTERVENTION(%) | COUNTERFACTUAL |
|---|---|---|---|
| PENDULUM | 57.1 | 39.6 | 87.7 |
| FLOW | 75.0 | 53.4 | 83.7 |
| CIRCUIT | 93.3 | 86.4 | 98.5 |
| OVERALL | 75.1 | 59.8 | 90.0 |

*Table 11.* **Variable-to-node matching frequency under GraphAlign-only training.** Without node-level alignment, variable identity remains unstable.

| | $\mathbf{U}_1^n$ | $\mathbf{U}_2^n$ | $\mathbf{U}_3^n$ |
|---|---|---|---|
| $\mathbf{Q}_1$ | 0.27 | 0.43 | 0.30 |
| $\mathbf{Q}_2$ | 0.38 | 0.31 | 0.31 |
| $\mathbf{Q}_3$ | 0.06 | 0.35 | 0.59 |

### F.1. Post-hoc Diagnostic for GraphAlign

To diagnose why GraphAlign alone can hurt downstream performance, we train BridgeVLM (+GraphAlign) and match each learned variable feature $\mathbf{Q}_i$ to node-description embeddings $\mathbf{U}_j^n$ using Hungarian assignment on the test set. Table 11 reports normalized matching frequency.

The matching distribution is diffuse for $\mathbf{Q}_1$ and $\mathbf{Q}_2$, indicating that GraphAlign constrains edges but does not reliably ground variable identity. This supports our explanation that oracle graph supervision can be misapplied when latent variables are not aligned to node semantics.

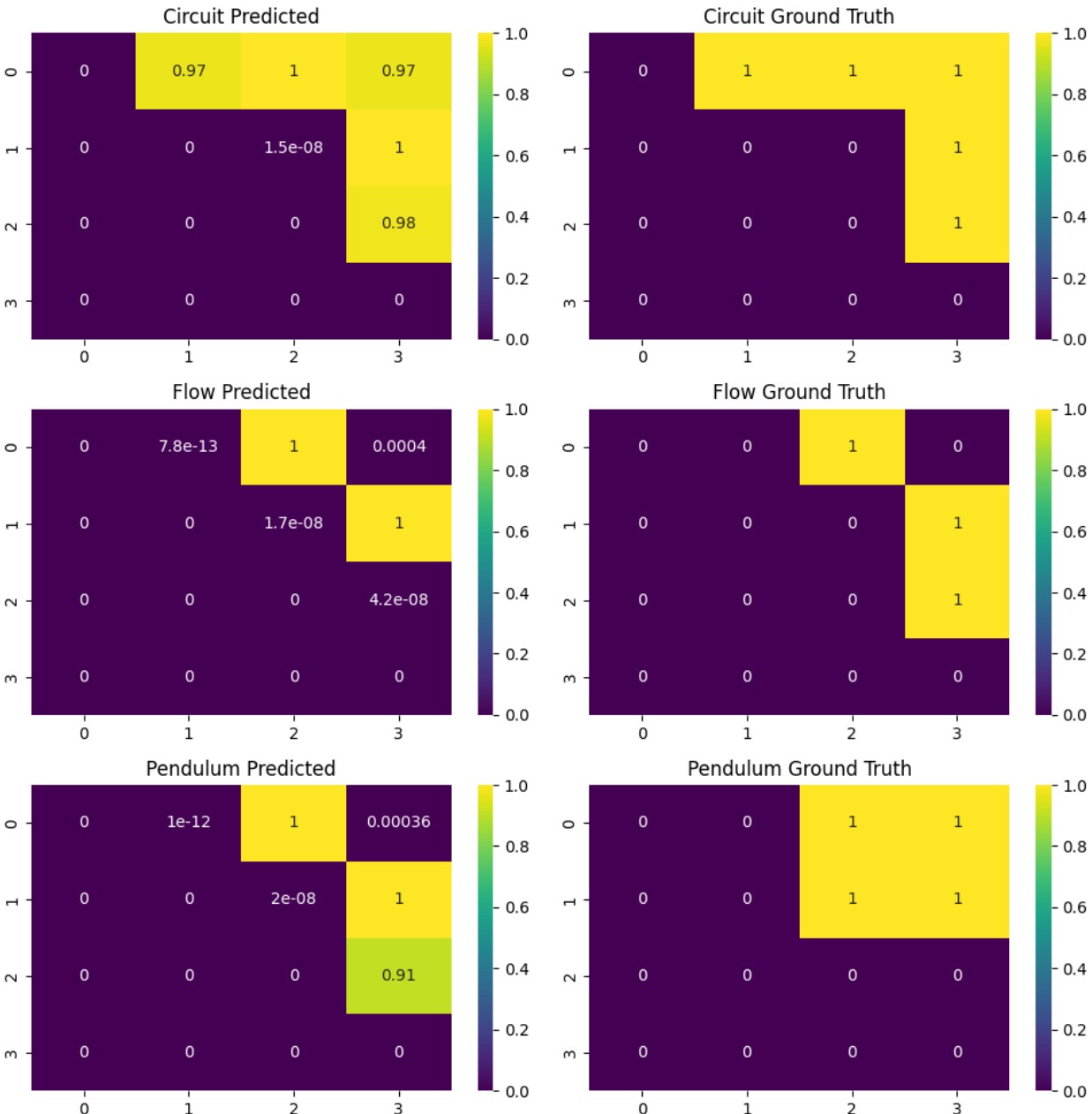

*Figure 5.* Visualization examples of induced DAG on CausalVLBench.

