# OpenReview forum: "From Prompts to Tokens: Internalizing Causal Supervision in Vision-Language Model for Multi-Image Causal Reasoning"
_ICML.cc/2026/Conference — ICML 2026 regular_

### Official Review · Reviewer_6Umn · 2026-03-06

**Soundness:** 2
**Presentation:** 3
**Significance:** 3
**Originality:** 3
**Overall Recommendation:** 4
**Confidence:** 3

**Summary:**

The paper proposes BridgeVLM, a framework that integrates causal supervision into the inner process of LVLMs for multi-image causal reasoning. It identifies two key disadvantages in the current textual prompt-based reasoning—interface gap (P1) and supervision gap (P2)—and addresses them by inducing a DAG from visual inputs, performing DAG-conditioned message propagation (RAMP) to produce Causal Tokens injected into the LLM decoder, and introducing M3S to handle heterogeneous and partial supervision signals. Experiments on Causal3D and CausalVLBench show substantial gains over prompt-level supervision baselines.

**Compliance With Llm Reviewing Policy:**

Affirmed.

**Final Justification:**

The paper integrates causal supervision into the inner process of LVLMs with a novel method. The experiment and rebuttal address most of my concerns.

**Key Questions For Authors:**

Q1: What is the time-cost during inference and training relative to the Phi-4 baseline?

Q2: Is there a more systematic analysis of why prompt-level causal injection sometimes degrades performance?

Q3: Is there any theoretical justification for using the unconditional Node→Graph→Node aggregation in Stage 3?

**Limitations:**

Yes

**Strengths And Weaknesses:**

**Strengths**

S1: The empirical demonstration that prompt-level causal injection can be ineffective or even harmful in visual causal reasoning provides a well-motivated starting point for the proposed approach.

S2: The ablation studies are well-designed.

S3: Strong empirical results with a 7B backbone, outperforming open-source baselines up to 32B and remaining competitive with a commercial model.

**Weaknesses**

W1: The paper adds multiple components but reports no training time, inference latency, or computational overhead compared to the base model.

W2: Generalization is undemonstrated. The authors acknowledge that evaluation assumes in-distribution test scenarios. Without any out-of-distribution experiment, it is unclear whether BridgeVLM learns transferable causal representations or overfits to dataset-specific patterns.

W3: All experiments use Phi-4-MMI-7B only. It is unclear whether the gains generalize across different LVLM architectures.

---

> ### Author Rebuttal · Authors · 2026-03-27
>
> We really appreciate your feedback.
>
> > **W1/Q1 (training/inference cost)**
> >
>
> We provide the full efficiency breakdown in our response to Reviewer RJWM, Q2. Briefly, under the same setting as Appendix B, BridgeVLM adds moderate overhead over Phi-4-MMI-7B: **+785M parameters, +102.32 ms training time/step, and +100.71 ms inference time/sample**. We view this as an efficiency–performance tradeoff: the method is not free, but the added cost is moderate relative to the gains on visual causal reasoning. Please note that our current implementation is unoptimized, so the more modest increases in parameters and FLOPs better reflect the architecture-level overhead (instead of latencies).
>
> > **W2 (OOD / transfer)**
> >
>
> We agree that the original paper does not establish strong OOD generalization. To probe transfer, we additionally evaluated the **CELLO intervention task**. Results (CoI / BAS / CDE / Avg) are:
>
> - **Phi-4-MMI (No-FT):** 60.0 / 60.0 / 78.5 / 66.2
> - **Phi-4-MMI (CELLO-adapted):** 80.0 / 76.0 / 95.2 / 83.7
> - **BridgeVLM (trained on CausalVLBench, zero-shot on CELLO):** 58.6 / 48.6 / 82.2 / 63.1
> - **BridgeVLM (CELLO-adapted):** 94.3 / 80.0 / 96.6 / 90.3
>
> Thus, we **do not** claim that zero-shot generalization to unseen causal scenes is solved (**63.1 vs. 66.2**). At the same time, we would like to clarify that formal visual causal reasoning is already **highly challenging even within a fixed scene family**: prior benchmarks consistently show substantial weaknesses of current LVLMs on intervention/counterfactual tasks. Our current contribution is primarily aimed at improving this already difficult setting.
> In addition, some visual causal benchmarks **intentionally include hypothetical scenes with artificially defined rules**, so cross-scene or cross-benchmark transfer often changes not only the image distribution, but also the variable semantics and underlying structural rules, which is **impossible to solve without extra information**. For this reason, we review such transfer as a particularly significant test. Correspondingly, after adaptation on CELLO, BridgeVLM reaches 90.3 average accuracy, outperforming the Phi-4 adaptation baseline (83.7) with low additional cost (~4 hours on 2xA100 for full CELLO finetuning). Dealing with OOD with zero-shot information for the real-world causal scenes is still our future work direction.
>
> > **W3 (single backbone)**
> >
>
> We agree that evaluating additional LVLM backbones would strengthen the paper, and we will do it in future work. In the paper, we intentionally fixed the backbone to Phi-4-MMI-7B to keep the comparison controlled and attribute gains to BridgeVLM rather than backbone differences. BridgeVLM is more like a lightweight add-on over visual embeddings plus decoder-side causal-token injection and does not rely on Phi-4-specific architectural changes; we therefore expect it to transfer to other decoder-based LVLMs.
>
> > **Q2: (why prompt-level causal injection can hurt)**
> >
>
> We do **not** claim that prompt-level causal injection is universally harmful. Its effect depends on the task and prompt form; e.g., CELLO shows that causal CoT can help in simpler/single-image settings. Our claim is narrower: in our multi-image setting, **long, highly structured causal prompts** can distract the model from the visual evidence. This conclusion is consistent with prior work showing that irrelevant textual context hurts reasoning (Shi et al., *Large Language Models Can Be Easily Distracted by Irrelevant Context*, ICML 2023) and that context length alone can degrade performance even with perfect image retrieval (Du et al., *Context Length Alone Hurts LLM Performance Despite Perfect Retrieval*, 2025). It is also consistent with the CausalVLBench paper. Our own ablation supports this: on **CausalVLBench intervention**, **Bridge (Answer)=35.6**, **+GraphPrompt=36.1**, but **+NodePrompt=28.3**. Since **NodePrompt** is much longer and contains full node explanations, while **GraphPrompt** is only a compact textual matrix, the degradation appears to come from **prompt length/format** rather than from causal information itself.
>
> > **Q3: (Node→Graph→Node in Stage 3)**
> >
>
> Stage 3 is intended to extract a **query-independent structured state** from the images, i.e., scene-level causal information, and pass it to the LLM decoder for downstream querying; this is why it is unconditional on the textual query. The **Node→Graph→Node** design serves one purpose:
>
> **Graph Tokens** act as a bottleneck that aggregates global information from Node Tokens and writes it back, so each final node contains both route-aware local information and graph-level context. We expect LVLM decoders to get the nodes and their related information (like their in/out edges) while querying Node Tokens.
>
> This intuition is related to **ISAB/Set Transformers** (Lee et al., 2019) and **Virtual Node** (Cai et al., PMLR 2023). Other designs could in principle serve a similar role if they satisfy the purpose.

---

> > ### Author Rebuttal · Reviewer_6Umn · 2026-04-02
> >
> > Thank you for the thorough explanation and experiment. The authors have addressed my major concerns.

---

> > > ### Author Response · Authors · 2026-04-08
> > >
> > > We sincerely thank you for your time, effort and insightful comments on our works!

---

### Official Review · Reviewer_iLKH · 2026-03-11

**Soundness:** 3
**Presentation:** 3
**Significance:** 3
**Originality:** 3
**Overall Recommendation:** 5
**Confidence:** 3

**Summary:**

The authors propose BridgeVLM, a vision-language model designed to improve causal reasoning by explicitly modeling latent variables and their relationships. The method introduces variable-level tokens extracted from visual features and a routing mechanism that induces a DAG representing dependencies between these variables. The model is trained using a supervision strategy that incorporates different forms of causal supervision (e.g., node descriptions and graph structures) when available. The goal is to enable the model to internalize structured causal representations while remaining compatible with standard VLM architectures.

The paper evaluates the approach on causal reasoning benchmarks (Causal3D and CausalVLBench), showing improvements over several LVLMs on tasks such as intervention and counterfactual reasoning. The authors also provide a number of ablations analyzing the effect of different supervision types and architectural components.

**Compliance With Llm Reviewing Policy:**

Affirmed.

**Final Justification:**

I am raising my score to a 5, since the authors addressed my main concerns and provided extra useful results and diagnostics during the rebuttal.

**Key Questions For Authors:**

1. The paper mainly compares against standard LVLMs. Have the authors considered comparing against approaches that explicitly construct structured scene representations (e.g., slot-based object-centric models, scene-graph + reasoning pipelines, or neural-symbolic VQA systems)? Such comparisons would help isolate whether the gains come from the proposed DAG-based routing or from introducing structured intermediate variables in general.

2. The evaluation focuses on Causal3D and CausalVLBench. Do the authors expect the proposed architecture to generalize to other benchmarks? Additional experiments (or discussion) would help support the broader claims about causal reasoning capabilities.

3. In line 378 the authors mention *TokenAlign*. Is this a typo? Can the authors explain the whole sentence, which seems contradictory to the sentence above? Also, first sentence of the *Clarification* paragraph (lines 234–242) is grammatically wrong and should be clarified.

**Limitations:**

Yes, but it should make clear that evaluation is currently limited to a small number of controlled causal benchmarks and that performance on broader real-world reasoning tasks remains to be demonstrated.

**Strengths And Weaknesses:**

**Strengths**
- The paper addresses an important problem: improving causal reasoning in VLMs. Explicitly incorporating structured causal representations is a promising direction.
- The proposed architecture is reasonably well motivated. The idea of learning variable-level tokens and inducing a DAG structure provides an interpretable intermediate representation and aligns well with causal reasoning frameworks.
- The experimental section is generally thorough. The authors perform several ablation studies analyzing the contribution of different supervision sources and architectural components, which helps understand the behavior of the method.
- The paper is overall clearly written and the narrative is easy to follow.

**Weaknesses**

- The main limitation is the evaluation and choice of baselines. The paper mainly compares against general-purpose LVLMs but does not include baselines that explicitly build structured scene representations, which are conceptually closer to the proposed approach. For example, object-centric models such as *Slot Attention* (Locatello et al., NeurIPS 2020), neural-symbolic pipelines like *NS-VQA* (Yi et al., CVPR 2018), or scene-graph reasoning systems such as *Graph R-CNN* (Yang et al., ECCV 2018). Including such comparisons would help clarify whether the gains come from the proposed routing/DAG mechanism or more generally from introducing structured intermediate representations.
- The evaluation is limited to Causal3D and CausalVLBench. Additional causal benchmarks like *CELLO* would help demonstrate that the approach generalizes beyond these datasets.
- Some aspects of the methodology would benefit from clearer analysis. For example, the discussion around graph supervision and alignment (e.g., the effect of GraphAlign) raises interesting hypotheses about node–variable misalignment but these claims are not empirically validated with targeted diagnostics.  Also, an interpretation/visualization of the learned variables (without supervision) would substantially strengthen the paper.

---

> ### Author Rebuttal · Authors · 2026-03-27
>
> Thank you for these suggestions. We address each point below.
>
> > **W1/Q1 (structured baselines).**
> >
>
> We agree that comparing only against general-purpose LVLMs does not fully isolate whether the gains come from structured intermediate representations in general or from our DAG/routing mechanism specifically. We therefore added a conceptually closer baseline that replaces our latent variable queries + DAG/RAMP + Graph Tokens with Slot Attention, and injects the resulting slot tokens directly into the decoder. This preserves structured intermediate representations but removes our causal routing.
>
> On CausalVLBench intervention, this baseline achieves:
> Pendulum 34.5 / Flow 43.3 / Circuit 37.0 / Avg 38.3.
>
> This is above our backbone baselines (31.0/33.2, depending on whether causal explanations are used as prompts), but remains well below BridgeVLM (54.4). Thus, **structured intermediate tokens do help, but they do not explain the full gain**; DAG-conditioned routing is a key contributor. In our view, the routing/DAG design not only provides structure but also aligns causal information more effectively with the Causal Tokens, where Node Tokens carries more precise node-specific causal information and Graph Tokens carry more significant global information.
>
> We did not use NS-VQA/scene-graph pipelines as primary baselines because their performance highly relies on external detectors/parsers and symbolic executors, which would substantially change the supervision setting.
>
> > **W2/Q2 (generalization to other benchmarks).**
> >
>
> We expect BridgeVLM to transfer to other benchmarks that require identifying variables and propagating relations, because it assumes only image(s) + query and does not require a benchmark-specific parser.
>
> That said, CELLO mainly focuses on **single-image causal reasoning**, which focuses on scene understanding, whereas our work is primarily motivated by extracting suitable causal relationships from **differences across multiple images**. Therefore, CELLO is not perfectly aligned with our target problem setting. Nevertheless, we additionally evaluated our work on the CELLO intervention task. Please see the detailed results in our response to Reviewer 6Umn, W2.
>
> The results indicate that BridgeVLM still improves over the backbone by +6.6 points there as well, suggesting that the learned Causal Tokens remain useful outside our main multi-image setting.
>
> > **W3 (diagnostics for GraphAlign / learned variables).**
> >
>
> Thank you for this suggestion. To address your concern, we added a lightweight post-hoc analysis. We trained **BridgeVLM (+GraphAlign)** on the CELLO intervention split, extracted the variable features $\mathbf{Q}$ on the test set, and then matched each $\mathbf{Q}_i$ to the corresponding node description embedding $\mathbf{U}^{n}_i$ using Hungarian assignment. We report the normalized matching frequency over the test set below:
>
> |  | $\mathbf{U}^{n}_1$ | $\mathbf{U}^{n}_2$ | $\mathbf{U}^{n}_3$ |
> | --- | --- | --- | --- |
> | $\mathbf{Q}_1$ | 0.27 | 0.43 | 0.30 |
> | $\mathbf{Q}_2$ | 0.38 | 0.31 | 0.31 |
> | $\mathbf{Q}_3$ | 0.06 | 0.35 | 0.59 |
>
> It shows that without node-level supervision, the model learns partially meaningful variable features, but **variable-to-node** correspondence remains unstable: $\mathbf{Q}_3$ shows a relatively clear preference for $\mathbf{U}^n_3$, whereas $\mathbf{Q}_1$ and $\mathbf{Q}_2$ remain much more diffuse. This supports our explanation that GraphAlign constrains edges but not variable identity; thus, message passing may still operate on misaligned nodes.
>
> > **Q3 (TokenAlign typo / clarification wording).**
> >
>
> Thank you for catching this. **TokenAlign** is a typo and should be **NodeAlign**.
>
> The intended meaning is: GraphAlign alone can be weak because oracle edges assume a consistent node ordering, while the latent variable features remain permutation-ambiguous without NodeAlign. RAMP follows the ordering induced by the induced DAG; if slot identities are misaligned, message passing is applied to the wrong nodes and can hurt performance. With NodeAlign, the slot semantics become anchored, so graph-based routing becomes effective.
>
> Regarding the **Clarification**, thank you as well for pointing out the grammatical issue. We have revised the sentence:
>
> “Although the routing DAG is directed (cause $\rightarrow$ effect), inference can still benefit from bidirectional information flow when identifying intervention targets and answering counterfactual queries, since these tasks often require the model to infer causes from observed effects."

---

> > ### Author Rebuttal · Reviewer_iLKH · 2026-04-03
> >
> > The authors have addressed my main concerns and provided further experimental justification.

---

> > > ### Author Response · Authors · 2026-04-08
> > >
> > > Thank you for your valuable feedback and for acknowledgment our response. We appreciate your time and effort.

---

### Official Review · Reviewer_RJWM · 2026-03-13

**Soundness:** 3
**Presentation:** 4
**Significance:** 3
**Originality:** 3
**Overall Recommendation:** 5
**Confidence:** 3

**Summary:**

This work proposes BridgeVLM to improve visual causal reasoning from prompting methods by introducing a model internal interface for causal message passing and M3S, a training interface for supervision from different granularities. More specifically, BridgeVLM extracts variable features from the images, induces a DAG over these variables to represent the causal relations, and applies route-aware message propagation to enable the information flow conditioned on the learned DAG. M3S is designed to supervise DAG under any available causal signals including node, edge, or global descriptions. Evaluations on causal benchmarks show improvement in performance over baseline conditions and baseline models, demonstrating the effectiveness of the proposed model architecture.

**Compliance With Llm Reviewing Policy:**

Affirmed.

**Final Justification:**

My final recommendation is accept, given this work is well-motivated, well-written, with sufficient empirical evidence to support the conclusions. The rebuttal addressed my original concern on the limitations, so I remain my positive assessment.

**Key Questions For Authors:**

Q1: from the ablation sections, there are multiple components that results in little or negative gain in performance, can the authors clarify a bit more on what are the main driving components, or what is a minimal subset that will provide a significant gain in the performance?

Q2: See W1, what is the relative efficiency/computation cost of the proposed method compared to baseline methods?

**Limitations:**

yes

**Strengths And Weaknesses:**

Strengths:

1. This work is well-motivated with the gap that causal signals are often passed in through prompting techniques and not directly injected into VLMs, and the proposed method provides a novel design to process causal information internally in the VLM.

2. The experiments are set up with multiple backbone variants and conditions, to clearly illustrate the performance gain of the proposed method. Claims on performance and model components are generally well supported by the main results as well as by the ablation results.

3. The presentation is well-structured with illustrative diagrams and highlighted intuitions and key takeaways.

Weaknesses:

1. There is limited discussion on whether there is additional computational overhead introduced by the proposed method compared to base models and conditions, is there an efficiency-performance tradeoff?

---

> ### Author Rebuttal · Authors · 2026-03-27
>
> > **Q1: There are multiple components that results in little or negative gain in performance, can the authors clarify a bit more on what are the main driving components, or what is a minimal subset that will provide a significant gain in the performance?**
> >
>
> Thanks for your question. We believe you are referring to two different ablation settings, and you see the negative gain from **Table 3**.
>
> **Table 2 (RQ2)** is the true **component ablation**. The **"None"** row corresponds to the full model, and other rows remove one component at a time. Removing any component causes a clear performance drop, indicating that all components are useful and necessary. Among them, the results show that **induced DAG** is one of the main structural drivers.
>
> We clarify that **Table 3** does **not** evaluate different architectural components. All components are kept fixed, and the table isolates the effect of different **supervision signals (RQ3/4)**. Therefore, the negative gain there should be interpreted as showing that some supervision signals are not suitable as standalone training objectives, rather than indicating that any component is ineffective.
>
> More specifically, **GraphAlign** alone leads to negative effect, which is supervising the induced DAG **only** with the **oracle causal graph** is actually detrimental to downstream inference.
> We believe this happens because:
> If the model is supervised only with the **oracle causal graph**, but without any **node-level supervision**, the variable features are learned in an essentially unsupervised manner. Even if these features capture the correct information, their correspondence to specific nodes is still not guaranteed; thus, their permutation still remains ambiguous. Therefore, without NodeAlign, GraphAlign may supervise mismatched variable identities. In other words, the oracle causal graph assumes a **fixed node ordering**. If the learned variable features do not follow the same ordering, message passing will be enforced over misaligned nodes, which can harm performance. This is illustrated in *Section 4.5* lines *371-379*.
>
> Therefore, our takeaway is that the **induced DAG component itself is most helpful, but supervision on this component alone is not**; grounding node semantics via **NodeAlign** is the main driver that makes the structural supervision effective.
>
> > **Q2/W1: What is the relative efficiency/computation cost of the proposed method compared to baseline methods? Is there an efficiency-performance tradeoff?**
> >
>
> Thank you for raising this question. We have measured the efficiency of BridgeVLM under the same setting as Appendix B with 2 NVIDIA A100 GPUs, and report the parameter count, per-forward FLOPs, and latencies below.
>
> |  | Phi-4-MMI-7B | BridgeVLM (Ours) |
> | --- | --- | --- |
> | # Params: Total | 5.574 B | 6.359 B |
> | # Params: Backbone (Include LoRA) | NA | 5.714 B |
> | # Params: Variable Features Encoder | NA | 226.68 M |
> | # Params: RAMP Module | NA | 188.80 M |
> | # Params: Causal Tokens Encoder | NA | 230.15 M |
> | Flops Per Forward | 11.713 T | 13.009 T |
> | Inference Latency Per Batch | 128.26 ms | 228.97 ms |
> | Training Latency Per Batch | 186.22 ms | 288.54 ms |
>
> As results show, there is indeed a limited efficiency-performance tradeoff: BridgeVLM requires a small number of additional parameters and computation time, but this overhead is accompanied by substantial gains in causal reasoning performance.
>
> Importantly, BridgeVLM uses the same Phi-4 backbone (Visual Encoder / LLM decoder); the additional latency comes from the newly introduced causal modules. The current FLOPS-latency gap is also **implementation-specific**: the Phi-4 backbone already benefits from mature **optimized kernels** (e.g., optimized attention paths such as FlashAttention2/SDPA, as well as fused projection/MLP implementations), while our added causal branch is currently implemented with **standard PyTorch operators** and has **not yet** been kernel-optimized. Therefore, the present latency numbers likely overestimate the intrinsic overhead of the method. We report them for transparency, while the architecture-level overhead is better reflected by the more modest increases in parameters and FLOPs.

---

> > ### Author Rebuttal · Reviewer_RJWM · 2026-04-03
> >
> > Thank you for the detailed explanations and efficiency reports, those have addressed my concerns.

---

> > > ### Author Response · Authors · 2026-04-08
> > >
> > > Thank you for your time and effort to review our paper and acknowledge our work.

---

### Decision · Program_Chairs · 2026-04-30

**Decision:**

Accept (regular)

**Comment:**

The paper proposes a vision-language model designed to improve causal reasoning by explicitly modeling latent variables and their relationships. It identifies two major disadvantages at supervision and inference times in the current textual prompt-based reasoning and addresses them by inducing a DAG from visual inputs, performing DAG-conditioned message propagation to produce Causal Tokens injected into the LLM decoder, and introducing M3S to handle heterogeneous and partial supervision signals. The paper received four reviews with there being several points of contention such as:

1. The question of generalization to other benchmarks than Causal3D and CausalVLBench to show the generalizability of the proposed approach.

2. A similar question was also asked regarding the use of different VLMs.

The rebuttal by the authors was detailed and reviewers were overall happy with the provided answers. During the reviewer discussions the decision converged to an acceptance. I concur with the reviewers and recommend accepatance. I request the authors to accomodate the rebuttal comments in the camera ready version of the paper.